# A Consistent and Differentiable
# $L_p$ Canonical Calibration Error Estimator

**Teodora Popordanoska**[*]
ESAT-PSI, KU Leuven
teodora.popordanoska@kuleuven.be

**Raphael Sayer**[*†]
University of Tübingen
raphael.sayer@uni-tuebingen.de

**Matthew B. Blaschko**
ESAT-PSI, KU Leuven
matthew.blaschko@esat.kuleuven.be

## Abstract

Calibrated probabilistic classifiers are models whose predicted probabilities can directly be interpreted as uncertainty estimates. It has been shown recently that deep neural networks are poorly calibrated and tend to output overconfident predictions. As a remedy, we propose a low-bias, trainable calibration error estimator based on Dirichlet kernel density estimates, which asymptotically converges to the true $L_p$ calibration error. This novel estimator enables us to tackle the strongest notion of multiclass calibration, called canonical (or distribution) calibration, while other common calibration methods are tractable only for top-label and marginal calibration. The computational complexity of our estimator is $\mathcal{O}(n^2)$, the convergence rate is $\mathcal{O}(n^{-1/2})$, and it is unbiased up to $\mathcal{O}(n^{-2})$, achieved by a geometric series debiasing scheme. In practice, this means that the estimator can be applied to small subsets of data, enabling efficient estimation and mini-batch updates. The proposed method has a natural choice of kernel, and can be used to generate consistent estimates of other quantities based on conditional expectation, such as the sharpness of a probabilistic classifier. Empirical results validate the correctness of our estimator, and demonstrate its utility in canonical calibration error estimation and calibration error regularized risk minimization.

## 1 Introduction

Deep neural networks have shown tremendous success in classification tasks, being regularly the best performing models in terms of accuracy. However, they are also known to make overconfident predictions [Guo et al., 2017], which is particularly problematic in safety-critical applications, such as medical diagnosis [Esteva et al., 2017, 2019] or autonomous driving [Caesar et al., 2020, Sun et al., 2020]. In many real world applications it is not only the predictive performance that is important, but also the trustworthiness of the prediction, i.e., we are interested in accurate predictions with robust uncertainty estimates. To this end, it is necessary that the models are uncertainty calibrated, which means that, for instance, among all cells that have been predicted with a probability of 0.8 to be cancerous, 80% should indeed belong to a malignant tumor.

The field of uncertainty calibration has been mostly focused on binary problems, often considering only the confidence score of the predicted class. However, this so called top-label (or confidence) calibration [Guo et al., 2017]) is often not sufficient in multiclass settings. A stronger notion of calibration is marginal (or class-wise) [Kull et al., 2019], that splits up the multiclass problem

---

[*]Equal contribution    [†]Most of this work was done while at KU Leuven

36th Conference on Neural Information Processing Systems (NeurIPS 2022).

Table 1: Properties of $ECE^{KDE}$ and other commonly used calibration error estimators.

| | Properties | | | |
|---|---|---|---|---|
| | Consistency | Scalability | De-biased | Differentiable |
| $ECE^{KDE}$ (**Our**) | ✓ | ✓ | ✓ | ✓ |
| $ECE^{bin}$ | ✗ [Vaicenavicius et al., 2019] | ✗ | ✓ [Roelofs et al., 2022] | ✗ |
| Mix-n-Match | ✓ [Zhang et al., 2020] | ✗ | ✗ | ✓ |
| MMCE | ✗ [Kumar et al., 2018] | ✓ | ✗ | ✓ |

into $K$ one-vs-all binary ones, and requires each to be calibrated according to the definition of binary calibration. The most strict notion of calibration, called canonical (or distribution) calibration [Bröcker, 2009, Kull and Flach, 2015, Vaicenavicius et al., 2019], requires the whole probability vector to be calibrated. The curse of dimensionality makes estimation of this form of calibration difficult, and current estimators, such as the binned estimator $ECE^{bin}$ [Naeini et al., 2015], MMCE [Kumar et al., 2018] and Mix-n-Match [Zhang et al., 2020], have computational or statistical limitations that prevent them from being successfully applied in this important setting. Specifically, the binned estimator is sensitive to the binning scheme and is asymptotically inconsistent in many situations [Vaicenavicius et al., 2019], MMCE is not a consistent estimator of $L_p$ calibration error, and Mix-n-Match, although consistent, is intractable in high dimensions and the authors did not implement it in more than one dimension.

We propose *a tractable, differentiable, and consistent estimator of the expected $L_p$ canonical calibration error*. In particular, we use kernel density estimates (KDEs) with a Beta kernel in binary classification tasks and a Dirichlet kernel in the multiclass setting, as these kernels are the natural choices to model densities over a probability simplex. In Table 1, we summarize and compare the properties of our $ECE^{KDE}$ estimator and other commonly used estimators. $ECE^{KDE}$ scales well to higher dimensions and it is able to capture canonical calibration with $\mathcal{O}(n^2)$ complexity.

Our contributions can be summarized as follows: 1. We develop a tractable estimator of canonical $L_p$ calibration error that is consistent and differentiable. 2. We demonstrate a natural choice of kernel. Due to the scaling properties of Dirichlet kernel density estimation, evaluating *canonical calibration* becomes feasible in cases that cannot be estimated using other methods. 3. We provide a second order debiasing scheme to further improve the convergence of the estimator. 4. We empirically evaluate the correctness of our estimator and demonstrate its utility in the task of calibration regularized risk minimization on various network architectures and several datasets.

## 2 Related Work

Calibration of probabilistic predictors has long been studied in many fields. This topic gained attention in the deep learning community since Guo et al. [2017] observed that modern neural networks are poorly calibrated and tend to give overconfident predictions due to overfitting on the NLL loss. The surge of interest resulted in many calibration strategies that can be split in two general categories, which we discuss subsequently.

**Post-hoc calibration strategies** learn a calibration map of the predictions from a trained predictor in a post-hoc manner, using a held-out calibration set. For instance, Platt scaling [Platt, 1999] fits a logistic regression model on top of the logit outputs of the model. A special case of Platt scaling that fits a single scalar, called temperature, has been popularized by Guo et al. [2017] as an accuracy-preserving, easy to implement and effective method to improve calibration. However, it has the undesired consequence that it clamps the high confidence scores of accurate predictions [Kumar et al., 2018]. Similar approaches for post-hoc calibration include histogram binning [Zadrozny and Elkan, 2001], isotonic regression [Zadrozny and Elkan, 2002], Bayesian binning into quantiles [Naeini and Cooper, 2016], Beta [Kull et al., 2017] and Dirichlet calibration [Kull et al., 2019]. Recently, Gupta et al. [2021] proposed a binning-free calibration measure based on the Kolmogorov-Smirnov test. In this approach, the recalibration function is obtained via spline-fitting, rather than minimizing a loss function on a calibration set. Ma et al. [2021] integrate ensemble-based and post-hoc calibration methods in an accuracy-perserving truth discovery framework. Zhao et al. [2021] introduce a new notion of calibration, called decision calibration, however, they do not propose an estimator of calibration error with statistical guarnatees.

**Trainable calibration strategies** integrate a differentiable calibration measure into the training objective. One of the earliest approaches is regularization by penalizing low entropy predictions [Pereyra et al., 2017]. Similarly to temperature scaling, it has been shown that entropy regularization needlessly suppresses high confidence scores of correct predictions [Kumar et al., 2018]. Another popular strategy is MMCE (Maximum Mean Calibration Error) [Kumar et al., 2018], where the entropy regularizer is replaced by a kernel-based surrogate for the calibration error that can be optimized alongside NLL. It has been shown that label smoothing [Szegedy et al., 2016, Müller et al., 2019], i.e. training models with a weighted mixture of the labels instead of one-hot vectors, also improves model calibration. Liang et al. [2020] propose to add the difference between predicted confidence and accuracy as auxiliary term to the cross-entropy loss. Focal loss [Mukhoti et al., 2020, Lin et al., 2017] has recently been *empirically* shown to produce better calibrated models than many of the alternatives, but does not estimate a clear quantity related to calibration error. Bohdal et al. [2021] derive a differentiable approximation to the commonly-used binned estimator of calibration error by computing differentiable approximations to the 0/1 loss and the binning operator. However, this approach does not eliminate the dependence on the binning scheme and it is not clear how it can be extended to calibration of the whole probability vector.

**Kernel density estimation** [Parzen, 1962, Rosenblatt, 1956, Silverman, 1986] is a non-parametric method to estimate a probability density function from a finite sample. Zhang et al. [2020] propose a KDE-based estimator of the calibration error (Mix-n-Match) for measuring calibration performance. Although they demonstrate consistency of the method, it requires a numerical integration step that is infeasible in high dimensions. In practice, they only implemented binary calibration, and not canonical calibration.

Although many calibration strategies have been empirically shown to decrease the calibration error, very few of them are based on an estimator of miscalibration. Our estimator is the first consistent, differentiable estimator with favourable scaling properties that has been successfully applied to the estimation of $L_p$ canonical calibration error in the multi-class setting.

## 3 Methods

We study a classical supervised classification problem. Let $(\Omega, \mathcal{A}, \mathbb{P})$ be a probability space, where $\Omega$ is the set of possible outcomes, $\mathcal{A} = \mathcal{A}(\Omega)$ is the sigma field of events and $\mathbb{P} : \mathcal{A} \to [0,1]$ is a probability measure, let $\mathcal{X} = \mathbb{R}^d$ and $\mathcal{Y} = \{1, ..., K\}$. Let $x : \Omega \to \mathcal{X}$ and $y : \Omega \to \mathcal{Y}$ be random variables, while realizations are denoted with subscripts. Suppose we have a model $f : \mathcal{X} \to \triangle^K$, where $\triangle^K$ denotes the $K - 1$ dimensional simplex as obtained, e.g., from the output of a final softmax layer in a neural network. We measure the (mis-)calibration of the model in terms of $L_p$ calibration error, defined below.

**Definition 3.1** (Calibration error, [Naeini et al., 2015, Kumar et al., 2019, Wenger et al., 2020]). The $L_p$ calibration error of $f$ is:

$$\mathrm{CE}_p(f) = \left( \mathbb{E}\left[ \left\| \mathbb{E}[y \mid f(x)] - f(x) \right\|_p^p \right] \right)^{\frac{1}{p}}. \tag{1}$$

We note that we consider multiclass calibration, and that $f(x)$ and the conditional expectation in Equation (1) therefore map to points on a probability simplex. We say that a classifier $f$ is perfectly calibrated if $\mathrm{CE}_p(f) = 0$.

In order to empirically compute the conditional expectation in Equation (1), we need to perform density estimation over the probability simplex. In a binary setting, this has traditionally been done with binned estimates [Naeini et al., 2015, Guo et al., 2017, Kumar et al., 2019]. However, this is not differentiable w.r.t. the function $f$, and cannot be incorporated into a gradient-based training procedure. Furthermore, binned estimates suffer from the curse of dimensionality and do not have a practical extension to multiclass settings. We consider an estimator for the $\mathrm{CE}_p$ based on Beta and Dirichlet kernel density estimates in the binary and multiclass setting, respectively. We require that this estimator is consistent and differentiable, such that we can train it in a calibration error regularized risk minimization framework. This estimator is given by:

$$\widehat{\mathrm{CE}_p(f)}^p = \frac{1}{n} \sum_{j=1}^n \left[ \left\| \widehat{\mathbb{E}[y \mid f(x)]} \Big|_{f(x_j)} - f(x_j) \right\|_p^p \right], \tag{2}$$

where $\mathbb{E}[\widehat{y \mid f(x)}]\Big|_{f(x_j)}$ denotes $\mathbb{E}[\widehat{y \mid f(x)}]$ evaluated at $f(x) = f(x_j)$. Let $p_{x,y}(x_i, y_i) = p_{y|x=x_i}(y_i)\, p_x(x_i)$ be the joint density. Then we define the estimator of the conditional expectation as follows:

$$\mathbb{E}[y \mid f(x)] = \sum_{y_k \in \mathcal{Y}} y_k\, p_{y|f(x)}(y_k) = \frac{\sum_{y_k \in \mathcal{Y}} y_k\, p_{f(x),y}(f(x), y_k)}{p_{f(x)}(f(x))}$$

$$\approx \frac{\sum_{i=1}^n k(f(x); f(x_i)) y_i}{\sum_{i=1}^n k(f(x); f(x_i))} =: \mathbb{E}[\widehat{y \mid f(x)}] \tag{3}$$

where $k$ is the kernel of a kernel density estimate evaluated at point $x$ and $p_{f(x)}$ is uniquely determined by $p_x$ and $f$.

**Proposition 3.2.** *Assuming that $p_{f(x)}(f(x))$ is Lipschitz continuous over the interior of the simplex, there exists a kernel $k$ such that $\mathbb{E}[\widehat{y \mid f(x)}]$ is a pointwise consistent estimator of $\mathbb{E}[y \mid f(x)]$, that is:*

$$\plim_{n \to \infty} \frac{\sum_{i=1}^n k(f(x); f(x_i)) y_i}{\sum_{i=1}^n k(f(x); f(x_i))} = \frac{\sum_{y_k \in \mathcal{Y}} y_k\, p_{f(x),y}(f(x), y_k)}{p_{f(x)}(f(x))}. \tag{4}$$

*Proof.* Let $k$ be a Dirichlet kernel [Ouimet and Tolosana-Delgado, 2022]. By the consistency of the Dirichlet kernel density estimators [Ouimet and Tolosana-Delgado, 2022, Theorem 4] Lipschitz continuity of the density over the simplex is a sufficient condition for uniform convergence of the kernel density estimate. This in turn implies that for a given $f$, for all $f(x) \in (0,1)$, $\frac{1}{n}\sum_{i=1}^n k(f(x); f(x_i)) y_i \xrightarrow{p} \sum_{y_k \in \mathcal{Y}} y_k\, p_{f(x),y}(f(x), y_k)$ and $\frac{1}{n}\sum_{i=1}^n k(f(x); f(x_i)) \xrightarrow{p} p_{f(x)}(f(x))$. Let $g(x) = 1/x$, then the set of discontinuities of $g$ applied to the denominator of the l.h.s. of (4) has measure zero since $\frac{1}{n}\sum_{i=1}^n k(f(x); f(x_i)) = 0$ with probability zero. From the continuous mapping theorem [Mann and Wald, 1943], it follows that $n/(\sum_{i=1}^n k(f(x); f(x_i))) \xrightarrow{p} 1/p_{f(x)}(f(x))$. Since products of convergent (in probability) sequences of random variables converge in probability to the product of their limits [Resnick, 2019], we have that $\sum_{i=1}^n k(f(x); f(x_i)) y_i\, g(\sum_{i=1}^n k(f(x); f(x_i))) \xrightarrow{p} \sum_{y_k \in \mathcal{Y}} y_k\, p_{f(x),y}(f(x), y_k)\, g(p_{f(x)}(f(x)))$, which is equal to the r.h.s. of (4). $\qquad\square$

The most commonly used loss functions are designed to achieve consistency in the sense of Bayes optimality under risk minimization, however, they do not guarantee calibration - neither for finite samples nor in the asymptotic limit. Since we are interested in models $f$ that are both accurate and calibrated, we consider the following optimization problem bounding the calibration error $\mathrm{CE}(f)$: $f = \arg\min_{f \in \mathcal{F}} \mathrm{Risk}(f), \text{s.t. } \mathrm{CE}(f) \leq B$ for some $B > 0$, and its associated Lagrangian:

$$f = \arg\min_{f \in \mathcal{F}} \Big( \mathrm{Risk}(f) + \lambda \cdot \mathrm{CE}(f) \Big). \tag{5}$$

**Mean squared error in binary classification** As a first instantiation of this framework we consider a binary classification setting, with mean squared error $\mathrm{MSE}(f) = \mathbb{E}[(f(x) - y)^2]$ as the risk function, jointly optimized with the $L_2$ calibration error $\mathrm{CE}_2$:

$$f = \arg\min_{f \in \mathcal{F}} \Big( \mathrm{MSE}(f) + \lambda\, \mathrm{CE}_2(f)^2 \Big) = \arg\min_{f \in \mathcal{F}} \Big( \mathrm{MSE}(f) + \gamma \mathbb{E}\Big[ \mathbb{E}[y \mid f(x)]^2 \Big] \Big) \tag{6}$$

where $\gamma = \frac{\lambda}{\lambda+1} \in [0,1]$. The full derivation using the MSE decomposition [Murphy, 1973, Degroot and Fienberg, 1983, Kuleshov and Liang, 2015, Nguyen and O'Connor, 2015] is given in Appendix A. For optimization we wish to find an estimator for $\mathbb{E}[\mathbb{E}[y \mid f(x)]^2]$. Building upon Equation (3), a partially debiased estimator can be written as:

$$\mathbb{E}\Big[\widehat{\mathbb{E}[y \mid f(x)]^2}\Big] \approx \frac{1}{n} \sum_{j=1}^n \frac{\left(\sum_{i \neq j} k(f(x_j); f(x_i)) y_i\right)^2 - \sum_{i \neq j} \left(k(f(x_j); f(x_i)) y_i\right)^2}{\left(\sum_{i \neq j} k(f(x_j); f(x_i))\right)^2 - \sum_{i \neq j} \left(k(f(x_j); f(x_i))\right)^2}. \tag{7}$$

Thus, the conditional expectation is estimated using a ratio of unbiased estimators of the square of a mean.

**Proposition 3.3.** *Equation* (7) *is a ratio of two U-statistics and has a bias converging as* $\mathcal{O}\left(\frac{1}{n}\right)$.

The proof is given in Appendix B.

**Proposition 3.4.** *There exist de-biasing schemes for the ratios in Equation* (7) *and Equation* (3) *that achieve an improved* $\mathcal{O}\left(\frac{1}{n^2}\right)$ *convergence of the bias.*

Proofs are given in Appendix C and D.

In a binary setting, the kernels $k(\cdot, \cdot)$ are Beta distributions defined as:

$$k_{\mathrm{B}}(f(x_j), f(x_i)) := f(x_j)^{\alpha_i - 1}(1 - f(x_j))^{\beta_i - 1}\frac{\Gamma(\alpha_i + \beta_i)}{\Gamma(\alpha_i)\,\Gamma(\beta_i)}, \tag{8}$$

with $\alpha_i = \frac{f(x_i)}{h} + 1$ and $\beta_i = \frac{1 - f(x_i)}{h} + 1$ [Chen, 1999, Bouezmarni and Rolin, 2003, Zhang and Karunamuni, 2010], where $h$ is a bandwidth parameter in the kernel density estimate that goes to 0 as $n \to \infty$. We note that the computational complexity of this estimator is $\mathcal{O}(n^2)$. If we would use this within a gradient descent training procedure, the density can be estimated using a mini-batch and therefore the $\mathcal{O}(n^2)$ complexity is w.r.t. the size of a mini-batch, not the entire dataset.

The estimator in Equation (7) is a ratio of two second order U-statistics that converge as $n^{-1/2}$ [Ferguson, 2005]. Therefore, the overall convergence will be $n^{-1/2}$. Empirical convergence rates are calculated in Appendix G and shown to be close to the theoretically expected value.

**Multiclass calibration with Dirichlet kernel density estimates** There are several definitions of multiclass calibration that vary in terms of how strictly they define the calibration of the probability vector $f(x)$. The strongest notion of multiclass calibration, and the one that we focus on in this paper, is canonical (also called multiclass or distribution) calibration [Bröcker, 2009, Kull and Flach, 2015, Vaicenavicius et al., 2019], which requires that the whole probability vector $f(x)$ is calibrated (Definition 3.1). Its estimator is:

$$\widehat{\mathrm{CE}_p(f)}^p = \frac{1}{n}\sum_{j=1}^{n}\left\|\frac{\sum_{i \neq j}k_{\mathrm{Dir}}(f(x_j); f(x_i))y_i}{\sum_{i \neq j}k_{\mathrm{Dir}}(f(x_j); f(x_i))} - f(x_j)\right\|_p^p \tag{9}$$

where $k_{\mathrm{Dir}}$ is a Dirichlet kernel defined as:

$$k_{Dir}(f(x_j), f(x_i)) = \frac{\Gamma(\sum_{k=1}^{K}\alpha_{ik})}{\prod_{k=1}^{K}\Gamma(\alpha_{ik})}\prod_{k=1}^{K}f(x_j)_k^{\alpha_{ik}-1} \tag{10}$$

with $\alpha_i = \frac{f(x_i)}{h} + 1$ [Ouimet and Tolosana-Delgado, 2022]. As before, the computational complexity is $\mathcal{O}(n^2)$, irrespective of $p$.

This estimator is differentiable and furthermore, the following proposition holds:

**Proposition 3.5.** *The Dirichlet kernel based* CE *estimator is consistent when* $p_{f(x)}(f(x))$ *is Lipschitz continuous:*

$$\operatorname*{plim}_{n \to \infty}\frac{1}{n}\sum_{j=1}^{n}\left\|\frac{\sum_{i \neq j}^{n}k_{\mathrm{Dir}}(f(x_j); f(x_i))y_i}{\sum_{i \neq j}^{n}k_{\mathrm{Dir}}(f(x_j); f(x_i))} - f(x_j)\right\|_p^p = \mathbb{E}\left[\left\|\mathbb{E}[y \mid f(x)] - f(x)\right\|_p^p\right]^p.$$

*Proof.* Dirichlet kernel estimators are consistent when the density is Lipschitz continuous over the simplex [Ouimet and Tolosana-Delgado, 2022, Theorem 4], consequently, by Proposition 3.2 the term inside the norm is consistent for any fixed $f(x_j)$ (note that summing over $i \neq j$ ensures that the ratio of the KDE's does not depend on the outer summation). Moreover, for any convergent sequence also the norm of that sequence converges to the norm of its limit. Ultimately, the outer sum is merely the sample mean of consistent summands, which again is consistent. □

With this development, we have for the first time a consistent, differentiable, and tractable estimator of $L_p$ canonical calibration error with $\mathcal{O}(n^2)$ computational cost and $\mathcal{O}(n^{-1/2})$ convergence rate, with a debiasing scheme that achieves $\mathcal{O}(n^{-2})$ bias for $p \in \{1, 2\}$.

# 4 Empirical validation of $ECE^{KDE}$

Accurately evaluating the calibration error is a crucial step towards designing trustworthy models that can be used in societally important settings. The most widely used metric for evaluating miscalibration, and the only other estimator that can be straightforwardly extended to measure canonical calibration, is the histogram-based estimator $ECE^{bin}$. However, as discussed in Vaicenavicius et al. [2019], Widmann et al. [2019], Ding et al. [2020], Ashukha et al. [2020], it has numerous flaws, such as: (i) it is sensitive to the binning scheme (ii) it is severely affected by the curse of dimensionality, as the number of bins grows exponentially with the number of classes (iii) it is asymptotically inconsistent in many cases.

To investigate its relationship with our estimator $ECE^{KDE}$, we first introduce an extension of the top-label binned estimator to the probability simplex in the three class setting. We start by partitioning the probability simplex into equally-sized, triangle-shaped bins and assign the probability scores to the corresponding bin, as shown in Figure 1a. Then, we define the binned estimate of canonical calibration error as follows: $\mathrm{CE}_p(f)^p \approx \mathbb{E}\left[\|H(f(x)) - f(x)\|_p^p\right] \approx \frac{1}{n}\sum_{i=1}^n \|H(f(x_i)) - f(x_i)\|_p^p$, where $H(f(x_i))$ is the histogram estimate, shown in Figure 1b. The surface of the corresponding Dirichlet KDE is presented in Figure 1c. See Appendix F for (i) an experiment investigating their relationship for the three types of calibration (top-label, marginal, canonical), and with varying number of points used for the estimation, and (ii) another example of the binned estimator and Dirichlet KDE on CIFAR-10.

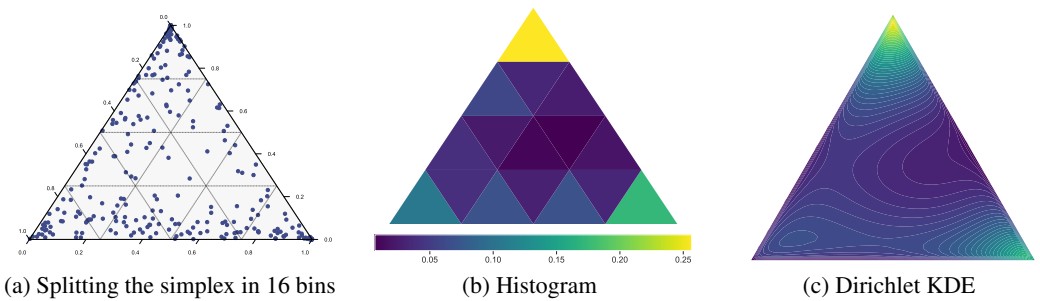

(a) Splitting the simplex in 16 bins     (b) Histogram     (c) Dirichlet KDE

Figure 1: Extension of the binned estimator $ECE^{bin}$ to the probability simplex, compared with $ECE^{KDE}$. The $ECE^{KDE}$ achieves a better approximation to the finite sample, and accurately models the fact that samples tend to be concentrated near low dimensional faces of the simplex.

**Synthetic experiments** We consider an extension of $ECE^{bin}$ to arbitrary number of classes and investigate its performance compared to $ECE^{KDE}$. Since on real data the ground truth calibration error is unknown, we generate synthetic data with known transformations, with the following procedure. First, we sample uniformly from the simplex using the Kraemer algorithm [Smith and Tromble, 2004]. Then, we apply temperature scaling with $t_1 = 0.6$ to simulate realistic scenarios where the probability scores are concentrated along lower dimensional faces of the simplex. We generate ground truth labels according to the sampled probabilities, and therefore, obtain a perfectly calibrated classifier. Subsequently, the classifier is miscalibrated by additional temperature scaling with $t_2 = 0.6$. Figure 2a depicts the performance of the two estimators as a function of the sample size on generated data for 4 and 8 classes. $ECE^{KDE}$ converges to the ground truth value obtained by integration in both cases, whereas $ECE^{bin}$ provides poor estimates even with 20000 points.
In another experiment with synthetic data we look at the bias of the sharpness[3] term in a binary setting. In Figure 2b we plot the estimated value of the sharpness term for varying number of samples, both using the partially debiased ratio from Equation (7), and the ratio debiased with the scheme introduced in Appendix D. A sigmoidal function is applied to the calibrated data to obtain an uncalibrated sample that is used to compute the partially debiased and the fully debiased ratio of the sharpness term. The ground truth value is obtained by using 100 million samples to compute the ratio with the partially debiased version, as it converges asymptotically to the true value due to its consistency. We use a

---

[3] The sharpness is defined as $\mathrm{Var}(\mathbb{E}[y \mid f(x)])$ [Kuleshov and Liang, 2015]. Here we neglect the term that does not depend on $f(x)$, and thereby refer to $\mathbb{E}\left[\mathbb{E}[y \mid f(x)]^2\right]$ as the sharpness.

bandwidth of 0.5 and average over 10000 repetitions for each number of samples that range from 32 to 16384. We fix the location of the KDE at $f(x_j) = 0.17$.

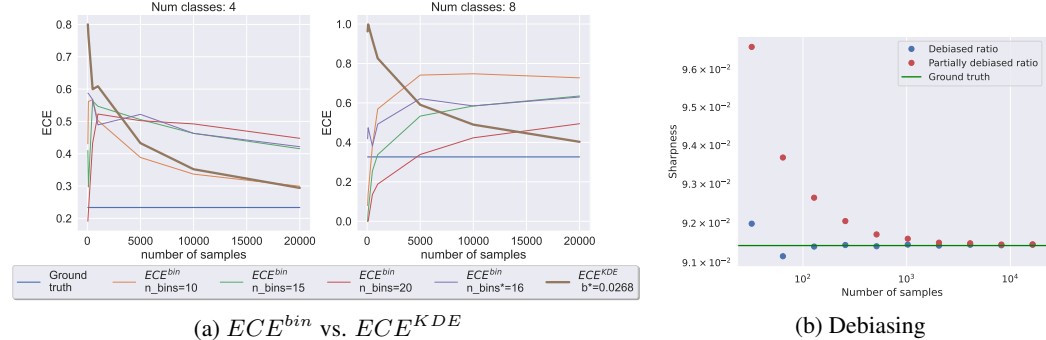

(a) $ECE^{bin}$ vs. $ECE^{KDE}$               (b) Debiasing

Figure 2: 2a Performance of $ECE^{bin}$ and $ECE^{KDE}$ on synthetic data for varying number of classes, as a function of the sample size. Ground truth represents the true value of the integral. $ECE^{bin}$ is calculated using several common choices for the number of bins (*n_bins* represents number of bins per-class.) *n_bins** and *b** are found as optimal values according to Doane's formula [Doane, 1976] and LOO MLE, respectively. $ECE^{KDE}$ converges to the true value in all settings, in contrast to $ECE^{bin}$. 2b Sharpness term evaluated for different numbers of samples with the partially debiased ratio from Equation (7), and with the debiasing scheme derived in Appendix D on synthetic data.

## 5 Calibration regularized training

**Empirical setup** To showcase our estimator in applications where canonical calibration is crucial, we consider two medical datasets, namely Kather and DermaMNIST. The Kather dataset [Kather et al., 2016] consists of 5000 histological images of human colorectal cancer and it has eight different classes of tissue. DermaMNIST [Yang et al., 2021] is a pre-processed version of the HAM10000 dataset [Tschandl et al., 2018], containing 10015 dermatoscopic images of skin lesions, categorized in seven classes. Both datasets have been collected in accordance with the Declaration of Helsinki. According to standard practice in related works, we trained ResNet [He et al., 2016], ResNet with stochastic depth (SD) [Huang et al., 2016], DenseNet [Huang et al., 2017] and WideResNet [Zagoruyko and Komodakis, 2016] networks also on CIFAR-10/100 [Krizhevsky, 2009]. We use 45000 images for training on the CIFAR datasets, 4000 for Kather and 7007 for DermaMNIST. The code is available at https://github.com/tpopordanoska/ece-kde.

**Baselines** *Cross-entropy*. The first baseline model is trained using cross-entropy (**XE**), with the data preprocessing, training procedure and hyperparameters described in the corresponding paper for the architecture.

*Trainable calibration strategies*. **KDE-XE** denotes the joint training of XE with our proposed estimator $ECE^{KDE}$, as defined in Equation (9). **MMCE** [Kumar et al., 2018] is a differentiable measure of calibration with a property that it is minimized at perfect calibration, i.e., MMCE is 0 if and only if $CE_p = 0$. It is used as a regulariser alongside NLL, with the strength of regularization parameterized by $\lambda$. **Focal loss (FL)** [Mukhoti et al., 2020] is an alternative to the cross-entropy loss, defined as $\mathcal{L}_f = -(1 - f(y|x))^\gamma \log(f(y|x))$, where $\gamma$ is a hyperparameter and $f(y|x)$ is the probability score that a neural network $f$ outputs for a class $y$ on an input $x$. Their best-performing approach is the sample-dependent FL-53, where $\gamma = 5$ for $f(y|x) \in [0, 0.2)$, and $\gamma = 3$ otherwise.

*Post-hoc calibration strategies*. Guo et al. [2017] investigated the performance of several post-hoc calibration methods and found **temperature scaling** to be a strong baseline, which we use as a representative of this group. It works by scaling the logits with a scalar $T > 0$, typically learned on a validation set by minimizing NLL. Following Kumar et al. [2018] and Mukhoti et al. [2020], we also use temperature scaling as a post-processing step for our method.

**Metrics** We report $L_1$ canonical calibration using our $ECE^{KDE}$ estimator, calculated according to Equation (9). Additional experiments with $L_1$ and $L_2$ top-label calibration on CIFAR-10/100 can be found in Appendix E.

**Hyperparameters** A crucial parameter for KDE is the bandwidth $b$, a positive number that defines the smoothness of the density plot. Poorly chosen bandwidth may lead to undersmoothing (small bandwidth) or oversmoothing (large bandwidth), as shown in Figure 3. A commonly used non-parametric bandwidth selector is maximum likelihood cross validation [Duin, 1976]. For our experiments we choose the bandwidth from a list of possible values by maximizing the leave-one-out likelihood (LOO MLE). The $\lambda$ parameter for weighting the calibration error w.r.t the loss is typically chosen via cross-validation or using a holdout validation set. We found that for KDE-XE, values of $\lambda \in [0.001, 0.2]$ provide a good trade-off in terms of accuracy and calibration error. The $p$ parameter is selected depending on the desired $L_p$ calibration error and the corresponding theoretical guarantees. The rest of the

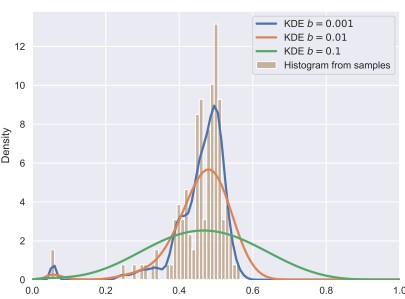

Figure 3: Effect of the bandwidth $b$ on the shape of the estimate.

hyperparameters for training are set as proposed in the corresponding papers for the architectures we benchmark. In particular, for the CIFAR-10/100 datasets we used a batch size of 64 for DenseNet and 128 for the other architectures. For the medical datasets, we used a batch size of 64, due to their smaller size.

## 5.1 Experiments

An important property of our $ECE^{KDE}$ estimator is differentiability, allowing it be used in a calibration regularized training framework. In this section, we benchmark KDE-XE with several baselines on medical diagnosis applications, where the calibration of the whole probability vector is of particular interest. For completeness, we also include an experiment on CIFAR-10.

Table 2 summarizes the canonical $L_1$ $ECE^{KDE}$ and Table 3 the accuracy, measured across multiple architectures. The bandwidth is chosen by LOO MLE. For MMCE and KDE-XE, we train the models with several values for the regularization weight, and report the best performing one. In Table 2 we notice that KDE-XE consistently achieves very competitive ECE values, while also boosting the accuracy, as shown in Table 3. Interestingly, we observe that temperature scaling does not improve canonical calibration error, contrary to its reported improvements on top-label calibration. This observation that temperature scaling is less effective for stronger notions of calibration is consistent with a similar finding in Kull et al. [2019], where the authors show that although the temperature-scaled model has well calibrated top-label confidence scores, the calibration error is much larger for class-wise calibration.

Table 2: Canonical $L_1$ $ECE^{KDE}$ ($\downarrow$) for different loss functions and architectures, both trained from scratch (Pre T) and after temperature scaling on a validation set (Post T). Best results across Pre T methods are marked in bold.

| Dataset | Model | XE | | MMCE | | FL-53 | | **KDE-XE (Our)** | |
|---------|-------|------|------|------|------|------|------|------|------|
| | | Pre T | Post T | Pre T | Post T | Pre T | Post T | Pre T | Post T |
| Kather | ResNet-110 | 0.335 | 0.304 | 0.343 | 0.300 | 0.325 | 0.248 | **0.311** | 0.289 |
| | ResNet-110 (SD) | 0.329 | 0.334 | 0.235 | 0.159 | 0.209 | 0.122 | **0.198** | 0.147 |
| | Wide-ResNet-28-10 | 0.177 | 0.259 | 0.201 | 0.241 | 0.270 | 0.328 | **0.162** | 0.212 |
| | DenseNet-40 | 0.244 | 0.251 | 0.159 | 0.218 | 0.165 | 0.207 | **0.114** | 0.154 |
| DermaMNIST | ResNet-110 | 0.579 | 0.602 | 0.575 | 0.603 | 0.684 | 0.618 | **0.467** | 0.516 |
| | ResNet-110 (SD) | 0.534 | 0.571 | 0.470 | 0.526 | 0.567 | 0.594 | **0.461** | 0.538 |
| | Wide-ResNet-28-10 | 0.546 | 0.599 | 0.470 | 0.512 | 0.623 | 0.608 | **0.455** | 0.599 |
| | DenseNet-40 | 0.573 | 0.578 | 0.514 | 0.558 | 0.577 | 0.557 | **0.366** | 0.418 |
| CIFAR-10 | ResNet-110 | 0.133 | 0.170 | 0.171 | 0.196 | 0.138 | 0.171 | **0.126** | 0.163 |
| | ResNet-110 (SD) | **0.132** | 0.172 | 0.164 | 0.203 | 0.156 | 0.201 | 0.178 | 0.223 |
| | Wide-ResNet-28-10 | 0.083 | 0.098 | 0.143 | 0.155 | 0.147 | 0.177 | **0.077** | 0.091 |
| | DenseNet-40 | 0.104 | 0.131 | 0.133 | 0.155 | **0.081** | 0.081 | 0.098 | 0.124 |

Table 3: Accuracy (↑) computed for different architectures. Best results are marked in bold.

| Dataset | Model | XE | MMCE | FL-53 | **KDE-XE (Our)** |
|---------|-------|-----|------|-------|------------------|
| Kather | ResNet-110 | 0.840 | **0.860** | 0.844 | **0.860** |
| | ResNet-110 (SD) | 0.870 | 0.900 | 0.885 | **0.914** |
| | Wide-ResNet-28-10 | **0.933** | 0.899 | 0.873 | 0.921 |
| | DenseNet-40 | 0.913 | 0.93 | 0.916 | **0.941** |
| DermaMNIST | ResNet-110 | 0.720 | 0.721 | 0.674 | **0.744** |
| | ResNet-110 (SD) | 0.743 | 0.753 | 0.689 | **0.764** |
| | Wide-ResNet-28-10 | 0.736 | 0.741 | 0.715 | **0.754** |
| | DenseNet-40 | 0.741 | **0.758** | 0.705 | 0.748 |
| CIFAR-10 | ResNet-110 | 0.925 | **0.929** | 0.922 | **0.929** |
| | ResNet-110 (SD) | **0.926** | 0.925 | 0.92 | 0.907 |
| | Wide-ResNet-28-10 | **0.954** | 0.947 | 0.936 | **0.954** |
| | DenseNet-40 | 0.947 | 0.944 | **0.948** | 0.947 |

Figure 4 shows the performance of several architectures and datasets in terms of accuracy and $L_1$ $ECE^{KDE}$ for various choices of the regularization parameter for MMCE and KDE-XE. The 95% confidence intervals for $ECE^{KDE}$ are calculated using 100 and 10 bootstrap samples on the medical datasets and CIFAR-10, respectively. In all settings, KDE-XE Pareto dominates the competitors, for several choices of $\lambda$. For example, on DermaMNIST trained with DenseNet, KDE-XE with $\lambda = 0.2$ reduces $ECE^{KDE}$ from 66% to 45%.

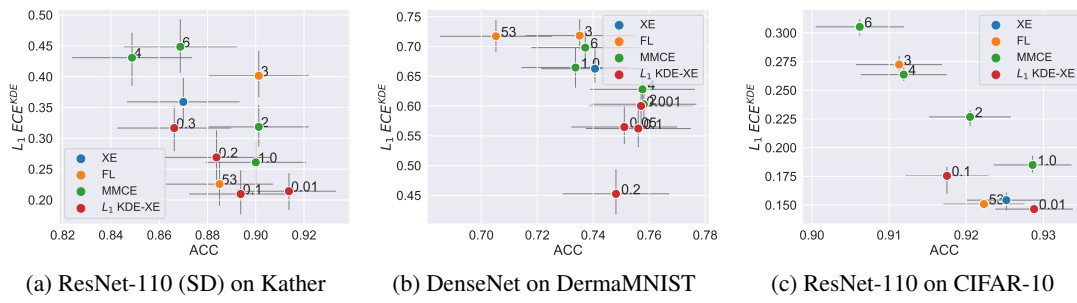

(a) ResNet-110 (SD) on Kather    (b) DenseNet on DermaMNIST    (c) ResNet-110 on CIFAR-10

Figure 4: Canonical calibration on various datasets and architectures. The numbers next to the points denote the value of the regularization parameter. KDE-XE outperforms the competitors, both in terms of accuracy and calibration error, for several choices of $\lambda$.

**Training time measurements**  In Table 4 we summarize the running time per epoch of the four architectures, with regularization (KDE-XE), and without regularization (XE). We observe only an insignificant impact on the training speed when using KDE-XE, dispelling any concerns w.r.t. the computational overhead.

To summarize, the experiments show that our estimator is consistently producing competitive calibration errors with other state-of-the-art approaches, while maintaining accuracy and keeping the computational complexity at $\mathcal{O}(n^2)$. We note that within the proposed calibration-regularized training framework, this complexity is w.r.t. to a mini-batch, and the added cost is less than a couple percent. Furthermore, the $\mathcal{O}(n^2)$ complexity shows up in other related works [Kumar et al., 2018, Zhang et al., 2020], and is intrinsic to the problem of density estimators of calibration error. As a future work, a larger scale benchmarking will be beneficial for exploring the limits of canonical calibration using Dirichlet kernels.

Table 4: Training time [sec] per epoch for XE and KDE-XE for different models on CIFAR-10.

| **Dataset** | **Model** | **XE** | **KDE-XE** |
|-------------|-----------|--------|------------|
| CIFAR-10 | ResNet-110 | 51.8 | 53.0 |
| | ResNet-110 (SD) | 45.0 | 46.0 |
| | Wide-ResNet-28-10 | 152.9 | 154.9 |
| | DenseNet-40 | 103.2 | 106.8 |

# 6    Conclusion

In this paper, we proposed a consistent and differentiable estimator of canonical $L_p$ calibration error using Dirichlet kernels. It has favorable computational and statistical properties, with a complexity of $\mathcal{O}(n^2)$, convergence of $\mathcal{O}(n^{-1/2})$, and a bias that converges as $\mathcal{O}(n^{-1})$, which can be further reduced to $\mathcal{O}(n^{-2})$ using our debiasing strategy. The $ECE^{KDE}$ can be directly optimized alongside any loss function in the existing batch stochastic gradient descent framework. Furthermore, we propose using it as a measure of the highest form of calibration, which requires the entire probability vector to be calibrated. To the best of our knowledge, this is the only metric that can tractably capture this type of calibration, which is crucial in safety-critical applications where downstream decisions are made based on the predicted probabilities. We showed empirically on a range of neural architectures and datasets that the performance of our estimator in terms of accuracy and calibration error is competitive against the current state-of-the-art, while having superior properties as a consistent estimator of canonical calibration error.

## Acknowledgments

This research received funding from the Research Foundation - Flanders (FWO) through project number S001421N, and the Flemish Government under the "Onderzoeksprogramma Artificiële Intelligentie (AI) Vlaanderen" programme. R.S. was supported in part by the Tübingen AI centre.

## Ethical statement

The paper is concerned with estimation of calibration error, a topic for which existing methods are deployed, albeit not typically for canonical calibration error in a multi-class setting. We therefore consider the ethical risks to be effectively the same as for any probabilistic classifier. Experiments apply the method to medical image classification, for which misinterpretation of benchmark results with respect to their clinical applicability has been highlighted as a risk, see e.g. Varoquaux and Cheplygina [2022].

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
