# OpenReview forum: "A Consistent and Differentiable Lp Canonical Calibration Error Estimator"
_NeurIPS.cc/2022/Conference — NeurIPS 2022 Accept_

### Official Review · Reviewer_GyKQ · 2022-07-08

**Rating:** 6
**Confidence:** 3
**Soundness:** 2 fair
**Presentation:** 2 fair
**Contribution:** 3 good

**Summary:**

Classifiers with deep neural networks tend to output overconfident class-posterior probabilities.
This paper proposes a novel method for calibrating the confidence (i.e., the predicted class-posterior probability) during training a classifier with deep neural networks. The proposed method adds a term to the loss for penalizing miscalibrated confidence scores.
The added term is an estimate of the $L_p$ distance between the predicted class-posterior probability and the true one, which is differentiable so that we can use the gradient descent method for training.
They show that the estimator of the discrepancy is statistically consistent, and calculation of it takes $O(B^2)$ time for mini-batch size $B$.
The paper presents experiments with a synthetic dataset and a three benchmark datasets whose results support the usefulness of the proposed method.

**Questions:**

- Denoting the mini-batch size as $B$, the overhead in the training time is indeed $O(B^2)$, but we only need to evaluate the function $B$ points in each iteration (which is nice). If this is correct, I suggest the authors mention this in the paper.
- I am not very sure if "scalable" is the right word for describing the overhead of $O(B^2)$. I suggest simply removing this from the title or using milder words.
- I request that the authors clearly present the mini-batch sizes used in the experiments rather than just referring to other papers.
- I do not think the statements about consistency in the paper are precise. The convergence has to be stated in terms of convergence in probability. I suggest making them more formal. The proofs are also very rough should be improved.
- Also, the paper only provides point-wise convergence analyses although the proposed method performs optimization over a function class. I suggest providing convergence analysis that holds uniformly over the function class.
- Are the color scales of Figure 1(b) and Figure 1(c) the same?
- "realistic scenarios where the probability scores are concentrated along lower dimensional faces of the simplex": What makes the authors believe it is "realistic"?
- How did the authors choose the bandwidth (0.5) and the location of KDE (0.17) for the synthetic data experiments?
- "We found that for KDE-XE, values of $\lambda \in [0.001, 0.2]$ provide a good trade-off in terms of accuracy and calibration error": How do we define a "good" trade-off?
- l.190: What does "high-cost settings" mean?
- What is the "partially" debiased estimator?

**Limitations:**

As I mentioned in "Questions" section, the training time overhead $O(B^2)$ can limit the mini-batch size and thus create bias. It would be nice if the authors could provide the maximum mini-batch size they can handle.

**Strengths And Weaknesses:**

# Strength
- The paper tackles the important task of confidence calibration for classification with deep neural networks.
- The authors provide some guarantees for their estimator of the miscalibration penalty term. Specifically, the estimator has $O(1/n)$ bias, and the rate of convergence is $O(1/n)$. The bias can be further improved to $O(1/n^2)$.
- The overhead in computation time added by the proposed method is $O(B^2)$ for mini-batch size $B$, which may be small when $B$ is small. In fact, they empirically showed that adding the proposed term only increases the training time by a few percentage points in their experiments.

# Weaknesses
- Their convergence analysis is done for a fixed function and is not guaranteed to hold for all functions of the feasible space of their optimization problem. Analysis of uniform convergence would be preferable.
- For a mini-batch size $B$, the estimator will have a bias of $O(1/B^2)$. For example, if $B$ is constant determined by the GPU memory size, we will have constant bias.
- Proofs in the paper are too brief and it is difficult to confirm the statements are correct.

---

> ### Author Response · Authors · 2022-08-02
> **Answer to Reviewer GyKQ (Part 1)**
>
> We thank the reviewer for the detailed feedback and concrete suggestions to improve our paper. Based on your summary of our paper, we would like to point out that the main contribution of the paper, as summarized by Reviewer Nx3X, is that we develop a consistent and differentiable *estimator* of calibration error. The estimator can be used independently of the calibration regularized training framework, for example, to evaluate the strongest notion of calibration - canonical. To the best of our knowledge, no other estimator exists with the same properties as ours. Due to its differentiability, it can additionally be used for training well-calibrated models as we have shown empirically, but that is not its only utility.
>
> **Convergence analysis**
>
> Indeed in certain settings it would be nice to have uniform convergence instead of point-wise. However, it would require us to limit our function class which in turn would limit the applicability of the resulting theoretical results. Moreover, proving uniform convergence for neural networks is still very much an open research field as it is non-trivial, and there is not a consensus on what function class is appropriate. We do agree that the characterization of function classes sufficient for uniform convergence is an interesting future line of research, and it could be that uniform convergence in this context could be shown for a relatively broad class of functions.
> Point-wise results are already useful in the setting of computing the calibration error of a fixed (set of) function(s), e.g. in the context of model selection or certification of model calibration properties.
> We note that focusing point-wise convergence allows us to leverage the theory of U-statistics without worrying about the function dependent constant which here gets abstracted.
>
> **Constant bias.**
>
> The batchsize B is a hyperparameter of our model and we fix it before training. Hence, it is true that the bias during training, which is $\mathcal{O}(\frac{1}{B^2})$, is constant. However, for any batch size used in practice this bias will be negligible. In particular, using batch sizes of 64 and 128 the bias is in $\mathcal{O}(\frac{1}{64^2})$ and $\mathcal{O}(\frac{1}{128^2})$, respectively, both of which are a small fraction of a percent and can be neglected for all practical purposes.
>
> **Proofs**
>
> We expanded the proof for Proposition 3.2, making it more formal by explicitly stating and citing the theorems used. We re-formulated the two propositions in terms of convergence in probability to make the statements about consistency more precise. The proof of Proposition 3.5 is relatively straightforward.  Proposition B.2 is a re-statement of a classical result in consistent notation, and we believe it is in sufficient detail.  Appendix B.2 is also in full detail.  Lemma C.1 and Appendices D & E follow the structure of other related ratio bias papers, which can be traced back to Tin (1965).  These are expanded to use more general U-statistics covariance results.  This strategy indeed may be worthy of a publication in the mathematical statistics community, but the principles are sound and our theoretical results are generalizations of known results on first-order ratio estimators.
>
> **Time complexity**
>
> We have to evaluate the KDE for each datapoint $f(x_i)$ at all other points $f(x_j) (i \neq j )$ of the batch of size B. Hence the time complexity of this computation is in $\mathcal{O}(B^2)$. However, the time complexity being quadratic in the number of samples is of no concern during training since it is w.r.t. to the batch size B and not w.r.t. to the entire training dataset. To this end, we have demonstrated that the overhead in computational time gained by adding the calibration regularization is insignificant (table 4).
>
> **Scalability**
>
> Thank you for the suggestion, we removed it from the title.
>
> **Mini-batch sizes**
>
> Following standard practice, on CIFAR-10/100 we used a batch size of 64 for DenseNet and 128 for the rest of the architectures, while on the medical datasets we used a batch size of 64 for all architectures, due to their smaller size. We added this information in the Hyperparameters paragraph in Section 5.

---

> > ### Author Response · Authors · 2022-08-02
> > **Answer to Reviewer GyKQ (Part 2)**
> >
> > **Color scales of Figure 1(b) and Figure 1(c)**
> >
> > Both subplots use the same color map from matplotlib ('viridis'). The color map underneath the histogram (1b) represents the fraction of points that fall into each bin (the points are shown in 1a), whereas 1c depicts the surface of Dirichlet KDE for the same points.
> >
> > **Probability scores concentrated along lower dimensional faces of the simplex**
> >
> > In a multi-class setting, by minimizing the cross-entropy loss, we train a classifier that for most samples will output high probability for one class, low probability for some classes and close to zero probability for the majority of the classes (e.g. two breeds of dog may be confused with each other but not with an aeroplane). Having classes with zero predicted probability corresponds to a problem of lower dimension. Therefore, in a realistic setting, i.e., in one where we use an actually useful classifier, the probability scores are concentrated along lower dimensional faces of the simplex.
> >
> > **Choice of bandwidth and location of KDE for synthetic experiments**
> >
> > The debiasing should hold for all bandwidths and all points. For the quantitative experiment we chose arbitrary points.
> >
> > **Trade-off**
> >
> > Generally speaking, some calibration strategies decrease the calibration error at the cost of a slight drop in accuracy. We consider a good trade-off the one where there is no (or very small) decrease in accuracy and a comparably large reduction in calibration error. In practice, which trade-off is considered preferable is application-dependent, i.e., sometimes one may wish to sacrifice a few accuracy points to get a nearly perfectly calibrated classifier. In our experiments, we found that a choice of $\lambda \in [0.001, 0.2]$ decreases the calibration error without substantially diminishing the accuracy.  Indeed, we sometimes noted an increase in accuracy for non-zero lambda.
> >
> > **l.190: What does "high-cost settings" mean?**
> >
> > Thanks for pointing out the ambiguity of that phrase. We changed it to “societally important”.
> >
> > **What is the "partially" debiased estimator?**
> >
> > We refer the reviewer to Equation 8, where we define the partially debiased estimator. We call it “partially” debiased because it has a debiased numerator and denominator, but the ratio of two unbiased estimators is not necessarily unbiased. Therefore, we derive a second-order debiasing procedure for the ratio in Appendix C and D.
> >
> > **Maximum mini-batch**
> >
> > In our experiments we used typical batch sizes of 64 and 128, however, one could easily increase the batch size to larger values if needed. Our estimator is not restricted to calibration regularized training, but rather it can also be used as a metric to evaluate calibration. We used it to evaluate canonical calibration on 10k points on the CIFAR test sets on a regular laptop with no problems.

---

> > > ### Comment · Reviewer_GyKQ · 2022-08-05
> > > **Additional suggestions**
> > >
> > > Thank you for the answers to the questions I raised in my review. I appreciate the update on Proposision 3.2.
> > > It would be even nicer if the authors could elaborate on which theorem of [Silverman, 1986, Chen, 1999, Ouimet and Tolosana-Delgado, 2022] is used in the proof of the proposition. Also, I suppose there are some assumptions, e.g., on the true probability density. If so, I request the authors clearly state them in the paper.

---

> > > > ### Author Response · Authors · 2022-08-08
> > > > **Thank you**
> > > >
> > > > Thank you for the great suggestions and the opportunity to make our propositions more precise. We updated the propositions and the proofs to explicitly state the theorems used and the assumptions on the probability density.

---

### Official Review · Reviewer_x5cZ · 2022-07-10

**Rating:** 6
**Confidence:** 3
**Soundness:** 3 good
**Presentation:** 4 excellent
**Contribution:** 3 good

**Summary:**

In this paper, the authors propose a novel strategy to estimate distributional or canonical calibration error based on Dirichlet kernel density estimates which asymptotically converges to the true Lp calibration error.  Such a calibration penalty is differentiable and can be incorporated while training a classifier.  The proposed approach improves the L1 ECE across multiple classification benchmarks and model architectures.

**Questions:**

How does adding the KDE calibration into the training pipeline explicitly control over-confidence in NN ? It would be beneficial to plot the accuracy vs confidence reliability plots along with comparisons with existing methods.

Can such a strategy be directly adopted for regression tasks ? It would be beneficial to demonstrate empirical results for regression.

Can this training strategy lead to models that can detect distribution shifts much better than other training strategies that include different calibration objectives ?  How does canonical calibration improve robustness to distribution shifts ?

Fig. 4 can be better expanded and scaled.

These are some of the typos I had spotted
Line 106 - It would be beneficial for readers if you can expand the notations used in the definition of the probability space (omega, A, P).
Line 149 - Need to specify that lambda and gamma are used interchangeably
Line 111: Left paranthesis can be removed before 'Calibration'
Eq. 12, the index j needs to be defined

Suggestions
It would be great if you can change some of the arxiv references to their actual conference/journal references (if published)


**Limitations:**

Yes

**Strengths And Weaknesses:**

Strengths

The idea of using Dirichlet kernel density estimates as the means to estimate canonical calibration is novel. The paper clearly illustrates how the conventional metrics such as ECE^BIN do not clearly convey the notion of canonical calibration and why it is important to estimate such a calibration error.  The work is mathematically well grounded and demonstrates a number of synthetic as well as real-data experiments. Overall the paper is well written with minor typos.

Weaknesses

Kindly refer to the questions below

---

> ### Author Response · Authors · 2022-08-02
> **Answer to Reviewer x5cZ**
>
> We thank the reviewer for the valuable feedback. We are pleased that the reviewer appreciates the novelty of our approach, finds the motivation and shortcomings of other approaches “clearly illustrated”, and considers our work “mathematically well grounded”. We address your questions below.
>
> **Controlling overconfidence in NN**
>
> One possible definition of overconfidence is that f(x) indicates a higher confidence (i.e. closer to 0 or 1) than E[y|f(x)].  This is one of the directions that is penalized by our minimization of an estimation of calibration error during training.  Indeed, it has been reported in a large number of publications that overconfidence is the most common form of miscalibration.
>
> **Reliability plots**
>
>
> Following your suggestion, in appendix F we included reliability plots for comparison with existing methods (XE, KDE-XE, MMCE and FL). As is standard practice, we plotted the reliability diagrams for top-label calibration on CIFAR-100 using ResNet and WideResNet.
>
> **Regression tasks**
>
> We thank the reviewer for the interesting suggestion. Indeed, calibration in regression problems is less studied, with most of the papers in the calibration literature focusing on classification. However, the definition of calibration in regression is different from the one in classification [see equation 3 and 4 in Kuleshov et al.], thus requiring more thorough analysis and adaptation of our estimator. We consider this a problem in its own right and therefore, out of the scope for this paper.
>
> *[Kuleshov et al.] “Accurate Uncertainties for Deep Learning Using Calibrated Regression”, Volodymyr Kuleshov, Nathan Fenner, Stefano Ermon, ICML 2018*
>
> **Distribution shift**
>
> The definition of calibration assumes that x, y are i.i.d realizations of the random variables X, Y ~ P, with P being the data distribution. Therefore, calibration on the i.i.d. validation dataset does not guarantee calibration under distributional shift for any method. With that being said, [Karandikar et al. NeurIPS 2021] demonstrate that under dataset shift, using calibration regularized training objectives result in better uncertainty estimates compared to post-hoc calibration methods. We believe that a study of different training objectives that incentivize calibration is a broad and interesting question, but it requires a systematic analysis across multiple datasets, different types of dataset shift, levels of corruption of the data, and methods of calibration. Therefore, we consider such analysis beyond the scope of this paper.
>
> *[Karandikar et al.,  NeurIPS 2021] “Soft Calibration Objectives for Neural Networks”, Archit Karandikar, Nicholas Cain, Dustin Tran, Balaji Lakshminarayanan, Jonathon Shlens, Michael C. Mozer, Becca Roelofs, NeurIPS 2021*
>
> **Typos and minor issues**
>
> Thank you very much for pointing these out. We corrected them in the updated manuscript. We also substituted arXiv references by their actual journal/conference reference, if available.

---

### Official Review · Reviewer_Nx3X · 2022-07-11

**Rating:** 8
**Confidence:** 2
**Soundness:** 3 good
**Presentation:** 3 good
**Contribution:** 4 excellent

**Summary:**

 The paper proposes a differentiable (and therefore trainable using gradient descent), low-bias calibration error estimator based on Dirichlet kernel density estimates, which is consistent w.r.t $L_{p}$ calibration error.

- The computational complexity of the estimator in the paper reported is $\mathcal{O}\left(n^{2}\right)$, while the convergence rate is $\mathcal{O}\left(n^{-1 / 2}\right)$, and it is unbiased up to $\mathcal{O}\left(n^{-2}\right)$, which the authors achieve by debiasing the estimate.

- This estimator shows strong performance in estimating the exact $L_1$, $L_p$ calibration error in a synthetic dataset with known calibrations. Since it is also a differentiable function, it can be used in conjunction with crossentropy loss to train neural nets, and it shows improved accuracy and calibration properties over other differentiable proxies.

**Questions:**

I mostly have minor nits to the paper, and I think it is a well-rounded paper in general, that covers most of my doubts.

1. "Interestingly, we observe that temperature scaling does not improve canonical calibration error, contrary to its reported improvements on top-label calibration." I think this line warrants much more study. Temperature scaling is a very commonly used technique to acheive quick calibration, and if it is not showing any improvements on the ECE metric proposed by the paper, that poses some interesting questions. Temperature scaling clearly improves model calibration, results in better performance across benchmarks on ECE (binned), Brier score, and OOD performance. However, this improvement in performance is not being captured by the new ECE metric? Do the authors have any intuition as to why? Is the KDE independent to division by a scalar term?

Minor nits:
- Line 81: "Maximum"

**Limitations:**

I believe the authors have addressed limitations well, and have a good section talking about potential societal impact.

**Strengths And Weaknesses:**

1. Originality

The field of uncertainty calibration has seen significant research in recent years in trying to find a differentiable approximation to calibration error, in order to be able to train calibration error using gradient descent. To my knowledge, previous methods have attempted to (1) perform soft binning, obtaining a differentiable approximation to the binned ECE (Bohdal et al 2021), (2) Mix-n-Match (Zhang et al 2020), which uses numerical integration and doesn't extend to multiclass calibration as far as I am aware. This paper addresses both these concerns quite well, as their metric is independent of binning, and doesn't require an expensive integration step.  I think the approach is quite original, and the authors have done a good job performing a literature survey. To my limited knowledge in this field, I didn't notice any similarities to previous work, and it seems like the current approach mitigates many concerns other methods have had.

2. Quality

I believe this paper is well written, has a clear motivation, strong experiments, and well-derived proofs and theorems. Not only does this paper propose a differentiable proxy to Lp calibration error with good guarantees, this proxy seems to provide good regularisation to empirical risk minimization, resulting in better classification performance on real-world datasets.

3. Clarity

I found the paper mostly easy to follow. I will admit that I found it a bit difficult to follow Appendix C, though this is due to my limited knowledge in proving such quantities. I feel like there wasn't much discussion about the bandwidth parameter $b$, and maybe explaining a little bit more how the leave-one-out MLE selection was performed would make this a bit clearer.

4. Significance

Model calibration is a very important field, and there has been much work on trying to find a differentiable proxy to ECE. To this extent, this paper accomplishes a significant milestone, as they propose a consistent, low-bias, differentiable approximator to Lp calibration error.

---

> ### Author Response · Authors · 2022-08-02
> **Answer to Reviewer Nx3X**
>
> We thank the reviewer for the comprehensive and detailed review that very well summarizes our motivation, the shortcomings of current methods and the improvements achieved by ours. This reaffirms that we managed to communicate our contributions clearly. Furthermore, we are glad that the reviewer appreciates that our approach “mitigates many concerns other methods have had”, and we are thrilled about the acknowledgement that the paper  “accomplishes a significant milestone“.
>
> **Temperature scaling.**
>
> We agree with the reviewer that studying the relation between top-label and canonical calibration is an interesting direction for future work. One of the main limitations of temperature scaling is that a single parameter is learned and it cannot be adjusted to act differently on different classes. While it has been shown to perform well for top-label calibration, other works have also reported a decrease in the performance for stronger notions of calibration. In particular, Figure 1 in [Kull et al. NeurIPS 2019]  depicts that even though temperature scaling works well for top-label (they call it confidence) calibration (subplot b), the calibration error is much larger when evaluated class-wise (subplot c). In appendix C1, they show that the same conclusion applies for many different calibration methods. To the best of our knowledge, no other paper evaluates canonical calibration and we hope that our estimator will enable further research in this direction.
>
> *[Kull et al. NeurIPS 2019] “Beyond temperature scaling: Obtaining well-calibrated multiclass probabilities with Dirichlet calibration”, Meelis Kull, Miquel Perello-Nieto, Markus Kängsepp, Telmo Silva Filho, Hao Song, Peter Flach, NeurIPS 2019*
>
> **LOO MLE procedure for bandwidth selection.**
>
> LOO MLE is a standard and commonly used strategy for bandwidth selection. As described in the Method section of [Duin 1976], we compute:
> $\max_h L(h) = \prod_{j=1}^n \hat{f_j}(x_j)$,
>
> where the leave-one-out estimator is given by [Equation 1.21, https://assets.press.princeton.edu/chapters/s8355.pdf ]:
>
> $\hat{f_j}(x_j) = \frac{1}{(n-1)h}\sum_{\substack{i=1, i \neq j}}^nk_{Dir}(f(x_j),f(x_i))$
>
> The implementation of this procedure can be found in the kde_ece.py file (line 210) in the code submitted as supplementary material.
>
> *[Duin 1976] On the choice of smoothing parameters for parzen estimators of probability density functions. Robert Duin, IEEE Transactions on Computers, 1976*

---

> > ### Comment · Reviewer_Nx3X · 2022-08-09
> > **Response to Rebuttal**
> >
> > I thank the authors for their response. I stand by my initial review and rating, I think this is a strong paper, and I thank them for the references they pointed me to.

---

### Author Response · Authors · 2022-08-02
**General response**

We thank all their reviewers for their time and efforts reviewing our paper. We appreciate the insightful feedback and constructive suggestions, which have helped us to improve the paper. The following revisions (changes in the main text are highlighted in red) were made to address the questions:

- We included reliability plots for top-label calibration on CIFAR-100 for each of the baselines in appendix F.
- We expanded the proof for Proposition 3.2 and re-formulated the two propositions in terms of convergence in probability.
- We added information about the batch sizes we used, we expanded the notation, improved the references, and fixed other smaller issues.

---

### Meta-Review · Area_Chair_74pD · 2022-08-24

**Recommendation:** Accept
**Confidence:** Certain

**Metareview:**

This meta review is based on the reviews, the authors rebuttal and the discussion with the reviewers, and ultimately my own judgement on the paper. There was a consensus that the paper contributes an interesting and novel strategy to model calibration based on Dirichelt kernel density estimates, and the reviewers praised several of its aspects. I feel this work deserves to be featured at NeurIPS and will attract interest from the community. I would like to personally invite the authors to carefully revise their manuscript to take into account the remarks and suggestions made by reviewers. Congratulations!

**Award:**

No

---

### Decision · Program_Chairs · 2022-09-14

Accept